# Consistency Checks for Language Model Forecasters

## Abstract

Forecasting is a task that is difficult to evaluate: the ground truth can only be known in the future. Recent work showing LLM forecasters rapidly approaching human-level performance begs the question: how to benchmark and evaluate those *instantaneously*? Following the consistency check framework, we measure forecasting performance on certain topics according to how consistent the predictions on different logically related questions are. The main consistency metric we use is one of *arbitrage*: for example, if a forecasting AI predicts 60% probability for both the Democratic and Republican parties to win the 2024 US presidential election, an arbitrageur could trade against the forecaster's predictions and make a profit. We build an automated evaluation system: starting from the instruction "query the forecaster's predictions on the topic of X", our evaluation system generates a set of base questions, instantiates the consistency checks from these questions, elicits the predictions of the forecaster, and measures the consistency of the predictions. We conclude with the possible applications of our work in steering and evaluating superhuman AI oracle systems.

## 1. Introduction

*Prediction markets* are markets that pay out contingent on an event. For a market such as "$1 if Jeb Bush is elected President in 2028", the price of it reflects the "market estimate" for the probability of that event. Prediction markets remain one of the most promising tools for aggregating information from disparate sources to arrive at "the most correct possible belief on a question after taking into account all relevant information" (Arrow et al., 2008; Hanson, 2002).

One way that AI could be used to great effect in the world is by making accurate forecasts on prediction markets, thereby providing human decision-makers with valuable information (or, alternatively by making bets on *decision markets*, see (Othman & Sandholm, 2010; Hanson, 2013; 1999)). An *AI oracle* that provides accurate predictions on which people then act upon is proposed as a *safe* way to use superhuman AI (Bengio, 2023). as opposed to agentic AI.

Until 2024, LLM forecasters generally performed poorly relative to human forecasters (Zou et al., 2022; Schoenegger & Park, 2023). However, two recent works have developed LLM-based forecasters that rival, and in some domains surpass crowds of human forecasters on forecasting websites such as Metaculus and Manifold Markets. Halawi et al. (2024) achieve this via retrieval- augmented generation (RAG), optimized prompting and fine-tuning, while (Schoenegger et al., 2024) achieve this via obtaining consensus from a crowd of LLM forecasters. There is also an ongoing "Humans vs Bots" competition underway on Manifold Markets (Manifold, 2024) [1]

A key question then becomes: *once LLM forecasters are better than human ones, how can we evaluate their predictions?* In particular, long-term forecasting questions are very important for decision-making, and finding ground truth for evaluation of superhuman models in such contexts is infeasible (Tetlock et al., 2024; Luke Muehlhauser, 2019).

One approach, spearheaded by Fluri et al. (2023), is that even when we cannot evaluate the *correctness* of LLM decisions, we can often evaluate their *logical consistency*. For example, if an LLM forecaster gives probabilities 0.5 and 0.7 to "Will Jeb be elected US president?" and "Will someone other than Jeb be elected US president?", this is necessarily inconsistent. That paper demonstrated that GPT-4 and GPT-3.5-turbo on their own were inconsistent quite often on simple consistency checks.

Our contributions in this work are:

- We produce a new *consistency evaluation* for LLM forecasters, consisting of tuples of binary forecasting questions which must satisfy some logical consistency rule. We autogenerate tuples of forecasting questions based either on a set of scraped forecasting questions, or just starting from a few keywords.

- We provide a principled metric, based on market arbi-

---

[1]Even current LLM forecasters may turn out to be valuable (Durland, 2024): one of the limitations of prediction markets is that they require subsidies from a market-maker, otherwise they are subject to no-trade theorems (Pennock & Sami, 2007). LLM-based forecasts are cheaper than human forecasters (under $1 per forecast for (Halawi et al., 2024), thus even a small market subsidy can elicit a very good estimate.

trage and logarithmic scoring, for measuring consistency violations of binary forecasts. This metric could also be used for training consistent forecasters.

- We generate LLM forecasts on a subset of our benchmark dataset, using both straightforward methods and the advanced setup from (Halawi et al., 2024), and evaluate them using our consistency violation metric.

*Notation.* Let $\mathrm{Prop}$ denote the set of forecasting questions we are interested in, $\Theta$ denote the set of possible outcomes/resolutions for an individual question, and $\mathrm{Forecast}$ denote the set of probability distributions on $\Theta$. A *Forecaster* is then a map $\mathbb{F} : \mathrm{Prop} \to \mathrm{Forecast}$. One special forecaster is the ground truth resolutions $\theta : \mathrm{Prop} \to \Theta$.

In this paper, we focus on $\mathrm{Prop}$ as a set of binary forecasting questions, so $\Theta = \{\top, \bot\}$ and $\mathrm{Forecast} = [0, 1]$. The type of optional resolutions $\Theta' := \Theta \cup \{\mathrm{None}\} = \{\top, \bot, \mathrm{None}\}$ will also be useful when considering conditional questions that may resolve to $\mathrm{None}$. Our methods could in principle be extended to study consistency between any types of probability distributions. We focus on binary questions following (Halawi et al., 2024), and because eliciting other probability distributions from LLMs is difficult.

## 2. Consistency checks

In line with (Fluri et al., 2023), a consistency check is conceptualized as a pair of n-ary relations $\mathcal{R} : \mathrm{Prop}^n \to \{\top, \bot\}$ and $\mathcal{S} : \mathrm{Forecast}^n \to \{\top, \bot\}$ and the predicate for forecasters $\mathbb{F}$ that $\mathcal{R}(x_1, \ldots x_n) \implies \mathcal{S}(\mathbb{F}(x_1), \ldots \mathbb{F}(x_n))$. In particular, this assertion must be satisfied by all feasible $\theta$, and also any "correct" forecasts generated by a world model that accurately accounts for aleatoric uncertainty. Violation of consistency is measured by some violation metric $\mathcal{V} : \mathrm{Forecast}^n \to \mathbb{R}$ which must satisfy $\mathcal{V}(\mathbb{F}(x_1), \ldots \mathbb{F}(x_n)) = 0 \iff \mathcal{S}(\mathbb{F}(x_1), \ldots \mathbb{F}(x_n))$. For example, the "negation" check NEGATION is given by the relations $\mathcal{R}(x_1, x_2) := x_1 = \neg x_2$ and $\mathcal{S}(x_1) + \mathcal{S}(x_2) = 1$. The full table of the consistency checks we use is given in Appendix A.

Improving upon (Fluri et al., 2023), we derive $\mathcal{V}$ (and therefore $\mathcal{S}$) from $\mathcal{R}$ in a principled way. We introduce two new *inconsistency metrics*: the *arbitrage metric* and the *frequentist metric* for measuring logical inconsistency in probabilistic forecasts.

The arbitrage metric is the minimum profit that an arbitrageur can be guaranteed making bets against the forecaster's predictions, under a proper scoring rule giving $\log(\mathbb{F}(x))$ profit for any true question outcome $x$. More precisely, suppose that the forecaster's probabilities $\mathbb{F}(x_1), \ldots \mathbb{F}(x_n)$ were prices offered by a market maker with market subsidy parameter \$1. If these probabilities

are inconsistent, then there are prices $p_1, \ldots p_n$ that an arbitrageur could bring the market prices to such that it is guaranteed to make a profit against the market maker no matter the outcome of each question. We set $\mathcal{V}(\mathbb{F}(x_1), \ldots \mathbb{F}(x_n))$ to be the maximum such "minimum profit" that the arbitrageur can guarantee by choosing appropriate $p_1, \ldots p_n$.

A more precise definition, including the description of the market maker and the proper scoring rule, is given in Appendix B. As an example, the arbitrage metric for the Negation Check can be derived exactly (Appendix B.2):

$$
\begin{aligned}
\mathcal{V}(\mathbb{F}(P), \mathbb{F}(\neg P)) = -2\log\Big( & \sqrt{\mathbb{F}(P)(1 - \mathbb{F}(\neg P))} \\
& + \sqrt{(1 - \mathbb{F}(P))\mathbb{F}(\neg P)} \Big)
\end{aligned}
$$

To illustrate: $\mathcal{V}(0.5, 0.6) \approx 0.01$, $\mathcal{V}(0.5, 0.51) \approx 10^{-4}$. The metric gets stricter for probabilities very close to 0 or 1. In our evals, for all types of checks, we say that a sampled check does not pass if $\mathcal{V} \geq 0.01$. We pick this threshold to corresponds to a clearly inconsistent world model: giving 110% probability in total to the events of Republican and Democratic parties winning the US presidential election.

We also compute a different, *frequentist* consistency metric. Consider a Monte Carlo forecaster that samples a world model $n$ times, and for any event, returns the fraction of samples in which the event occurs. The frequentist metric is the number of standard deviations a given tuple forecast is off from the mean Monte Carlo forecast, scaled to be independent of $n$. The full description is given in Appendix C.

## 3. Pipeline overview

A brief description of our overall pipeline for generating and evaluating consistency checks on forecasters:

- $(\cdots \longrightarrow P)$ A dataset of **base questions** is prepared from a combination of *(a)* scraping from online platforms such as Manifold and Metaculus and *(b)* synthetic generation.

- $(P \longrightarrow (P, Q))$ The base questions are synthetically **instantiated into tuples** that must satisfy certain consistency checks. E.g. every single base question $P$ is instantiated into a tuple $(P, \neg P)$; pairs of separate base questions $P, Q$ are instantiated into tuples like $(P, Q, P \land Q, P \lor Q)$.

- $((P, Q) \xrightarrow{\mathbb{F}} (p, q))$ The forecaster is separately queried to elicit **forecasts** on each base question, resulting in forecast tuples that should, if the forecaster is consistent, satisfy consistency properties. For example, in the case of $Q = \neg P$, it should satisfy $p + q = 1$.

- $((p,q) \xrightarrow{\mathcal{V}} \mathcal{V}(p,q))$ We score each tuple of forecasts for consistency with our violation metric.

$$\ldots \xrightarrow[\text{+scraping}]{\text{synthetic}} P \xrightarrow[\text{instantiation}]{\text{synthetic}} (P,Q) \xrightarrow{\mathbb{F}} (p,q) \xrightarrow{\mathcal{V}} \mathcal{V}(p,q)$$

Some details for each step are given below. Examples of data at each step of the pipeline are given in Appendix E.1; prompts for LLM calls used in each step before forecasting are given in Appendix E.2; the models used for each step in Appendix E.3.

### 3.1. Generating and scraping forecasting questions

**Real prediction market questions**   We extract an initial dataset of real-world questions from several prediction market platforms: PredictIt, Manifold Markets, and Metaculus. This yielded approximately 10,000 entries that encompassed various types of questions, including binary questions, multiple choice questions, opinion questions, and continuous value questions. These questions are useful both for testing and for generating new synthetic questions. For testing, we limit our sample to approximately 500 binary questions sourced from Metaculus. In particular, Metaculus has an advantage of providing clear ancillary details such as precise resolution criteria and background information. We conduct further processing to standardize the data and fill in any additionally needed information. An example of a processed question, including its relevant details, is provided in Appendix E.1. Although not used for this project, future work may consider using a broader source and other types of real prediction market questions.

**Synthetic question generation**   We generate questions by few-shot prompting, starting from a handpicked set of a few question titles for several topics. Once a question is verified to be valid, we add it to our question database, and reuse it in few-shot prompting for new questions. To get a more diverse set of questions, we additionally deduplicate the question database based on a similarity threshold on the `text-embedding-3-small` embeddings from OpenAI.

Once titles are generated, we generate question bodies and resolution dates using a few-shot prompt to `gpt-4o`. More details are in Appendix E.2.

**Verification and improvement from human feedback**
To improve the quality of synthetically generated questions, we developed a feedback form for human reviewers to evaluate and recommend modifications to the generated questions. Within this form, a human reviewer assesses a synthetically generated question and provides feedback on ambiguities,

any explicit errors, and the overall quality of the question. For instance, the reviewer may provide insights on ambiguity in resolution, suggest clarifications or deletions of specific details provided, identify incorrect resolution dates, or correct background information, among other aspects. We then use this feedback is to guide the LLM generation of a new set of questions. An example of the feedback form input can be found in Appendix F.

### 3.2. Instantiating consistency checks

The scraped base questions were subsequently used to synthetically generate tuples of logically related questions. For example, a pair of base questions $(P,Q)$ can be used to generate a 4-tuple $(P, Q, P \land Q, P \lor Q)$ for AndOrChecker, or a 3-tuple $(P, Q \land \neg P, P \lor Q)$ for ButChecker (see Appendix A for details). The main question content (titles and bodies) were generated synthetically (using `gpt-4o`), while the resolution dates and other properties were calculated systematically (e.g. the `max` of the resolution dates of the base questions).

We then conduct two measures to ensure the instantiated tuples are correct and sensical: relevance scoring, and verification that the tuples of questions indeed describe logically related events.

**Relevance scoring.**   When combining base questions into tuples, we have to take care to avoid off-distribution questions like "Is SpaceX going to be worth \$200B by 2030, given that Sri Lanka's rice production grows 40% by 2040?". For tuples instantiated from more than one base question, we sort 1000 potential base question combinations by their "relevance score" (obtained by querying an LLM and asking it to score how relevant the questions are to one another) and choose the top 50 for each consistency check (30 for synthetic base questions).

**Verification.**   The instantiated tuples of questions are then passed to another LLM call to reject if they do not fit their intended structure; for example, we detect if the resolution criteria of the second question are not truly a negation of the resolution criteria of the first question). We additionally inspect and filter a sample of instantiated tuples manually. For all checks except CONSEQUENCE, manual filtering did not remove any ambiguous or wrongly instantiated tuples; we conclude the failure rate of our instantiation is less than 5% on our base question distribution. For CONSEQUENCE, even after verification, our instantiation still produces ambiguous or invalid tuples over 30% of the time. For this check only, we generate 100 tuples from each of the scraped synthetic base question, and manually filter to 50/30 valid tuples.

### 3.3. Eliciting forecasts

In (Halawi et al., 2024), the authors created a system that can make forecasts on prediction market questions that are near or even exceed human levels. To achieve this, they developed a pipeline through which language models generate search queries from the forecast questions, retrieve news articles, and rate the relevance of said articles through a process of self fine-tuning to generate predictions.

We modify their setup to use `gpt-4o` instead of `gpt-4`, and use a slightly inferior RAG system (Google News instead of Newscatcher). Running this forecaster costs about $0.2 per question depending on the retrieved articles, or on the order of $0.5 per consistency check tuple.

## 4. Results

We evaluated the following forecasters:

- The state-of-the-art *Advanced Forecaster* set-up from (Halawi et al., 2024), except with `gpt-4o` for all model calls, and GNews instead of NewsCatcher.

- `gpt-4o` in a *Basic* setup (i.e. prompted directly, with no retrieval-augmented generation nor chain-of-thought).

- `gpt-3.5-turbo` in a *Basic* setup.

Our results for the Advanced Forecaster are reported in Table 1; results for the remaining forecasters are relegated to Appendix D. The #violations column counts the number of tuples for which the violation exceeded a certain threshold. For the arbitrage metric, this threshold is $10^{-2}$; for the frequentist metric, this is determined by rejecting the null hypothesis of a Monte Carlo forecaster, at significance level $p < 0.01$. The full exposition of the frequentist metric is in Appendix C. The arbitrage and frequentist metrics are not directly comparable, but the respective violation counts are: the hyperparameters for the violation threshold for the frequentist metric are tuned to have the same threshold as the arbitrage metric for NEGATION.

In any case, our sampling of consistency checks can discover inconsistencies in both pure and RAG-augmented LLM forecasters around half of the time, for most of the consistency checks in Table 2. Notably, the impact of introducing RAG and reasoning as in the Advanced Forecaster seems to be marginal at best, and even negative on some checks such as NEGATION. This is despite the overall higher Brier score of the Advanced Forecaster (Halawi et al., 2024), which suggests that there remains room to squeeze out further accuracy gains by improving inconsistency.

## 5. Related work

**Metamorphic and consistency checks.** Checking logical properties of outputs of programs under semantic-preserving transforms has a long history (Chen et al., 1998). Before (Fluri et al., 2023), variants of the consistency check framework were used for simple ML models (Christakis et al., 2022; Sharma & Wehrheim, 2020), vision (Hendrycks & Dietterich, 2019) and chat LLMs (Jang & Lukasiewicz, 2023), among other areas.

**Market-based approaches to AI reasoning and safety** Our focus on consistency may be viewed in the light of *market-based AIs*, i.e. AI frameworks that use an internal market to form beliefs and make decisions.

Most work on such frameworks is quite dated and has focused on agentic AIs in a reinforcement learning setting[2]. One recent work in which AIs only give probabilities to propositions is the *logical induction* framework (Garrabrant et al., 2020), which implements a prediction market for mathematical sentences to determine their probabilities. The market-based logical inductor can in fact be shown to be *unique* in satisfying a certain "generalized No Dutch Book Criterion" (Wentworth, 2019).

We have three main takeaways from this.

1. If our position here that consistency is a desirable feature for AI safety – or that it is an indicator of truth – is right, then this should be taken as a **point in favour of market-based AI frameworks**. Indeed, our arbitrage metric may be taken as a measure of how far a forecaster is from being a market, or a metric of bounded rationality.

2. (Garrabrant et al., 2020) does not use a static list of consistency checks, but instead it is designed to satisfy the "Generalized No Dutch Book Criterion" which says that *no polynomial-time trader can indefinitely make a profit off the market-maker*. Similarly, we may try to **dynamically generate consistency checks**, by having agents competitively try and come up with bets to profit off each other.

3. More generally, a **virtual market of AI forecasters**[3], can itself be regarded as an AI oracle. We may also study if such a market comprised of relatively primitive LLMs may outperform more advanced forecasters on consistency or ground-truth accuracy.

---

[2]e.g. the *learning classifier systems* framework pioneered by (Holland, 1986), with important later improvements from (Baum, 1999; Kwee et al., 2001; Chang et al., 2020)

[3]This might follow, for instance, the "Information Bazaar" framework implemented in (Rahaman et al., 2024)

*Table 1.* Consistency metrics for Advanced Forecaster from (Halawi et al., 2024).

| Checker | $n$ | Arbitrage metric | | | Frequentist metric | | |
|---|---|---|---|---|---|---|---|
| | | # violations | Violation (mean) | Violation (median) | # violations | Violation (mean) | Violation (median) |
| NEGATION | 80 | 42 | 0.0811 | 0.0120 | 43 | 0.2684 | 0.1550 |
| ANDOR | 80 | 47 | 0.0572 | 0.0139 | 52 | 0.2496 | 0.1741 |
| BUT | 80 | 61 | 0.0823 | 0.0441 | 64 | 0.3404 | 0.2939 |
| AND | 80 | 10 | 0.0130 | 0.0000 | 12 | 0.0582 | 0.0000 |
| OR | 80 | 26 | 0.0428 | 0.0012 | 28 | 0.1563 | 0.0708 |
| COND | 80 | 35 | 0.0288 | 0.0063 | 31 | 0.1252 | 0.0809 |
| CONDCOND | 80 | 25 | 0.0150 | 0.0000 | 32 | 0.1251 | 0.0836 |
| CONSEQUENCE | 80 | 4 | 0.0479 | 0.0000 | 4 | 0.0873 | 0.0000 |
| PARAPHRASE | 80 | 30 | 0.0100 | 0.0044 | 32 | 0.1032 | 0.0931 |

**Scalable oversight and failures of superhuman AI** The difficulty of evaluating models with superhuman performance in domains without a source of ground truth has long been acknowledged, and falls under the umbrella of *scalable oversight* (Amodei et al., 2016). Forecasting using AI oracles is one such domain. In addition to (Fluri et al., 2023), the use of consistency checks for scalable oversight has been studied in the context of AlphaZero (Lan et al., 2022) and in general question-answering tasks via debate (Irving et al., 2018).

**Consistency evaluations for LLMs.** Even on tasks where the ground truth is in principle knowable, consistency evaluations have long helped in cases where checking consistency is easier than getting the ground truth labels (Elazar et al., 2021). Li et al. (2023) check consistency of LLMs are *generators* ("Write a sentence funnier than sentence X") and *validators* ("Is sentence X or Y funnier?"). This paradigm is applicable for a general AI system forecasting: we can use "Give me an event which is more likely to happen than event X" as a generator, and check the probabilities afterward.

## 6. Future work

We believe that our consistency evaluation can be used to verify LLM truthfulness and reliability, particularly in light of the success of methods such as Contrast-Consistent Search (Burns et al., 2022; Kaarel et al., 2023), which detects a model's internal conception of truth based on its consistency properties. We foresee several promising directions:

**Consistency in decision-making.** AI systems may be used not only to make predictions/forecasts that inform decisions, but also to take decisions directly. Here too, we can have a notion of inconsistency: for example, *intransitive prefer-*

*ences* [4] – and analogously, an inconsistent decision-maker may be money-pumped by an arbitrageur.

We can think of an AI decision-maker as giving utility estimates to different options (and choosing the highest one), much like an AI forecaster gives probability estimates to different outcomes. However, the analogy is not completely straightforward, as AI decision makers may also differ on which options they list in their analysis.

**Training for consistency.** Modulo consideration of the cost-benefit to safety, our methods could be used train LLMs for consistency, with our violation metrics as a loss function. Future work could then look at how this impacts overall forecasting performance and other AI capabilities, as well as measure if this improves truthfulness overall, e.g. by comparing answers to the model's internal beliefs.

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

# A. Table of consistency checks

Table 2 gives a table consisting of all the consistency checks tested for in our benchmark. In most of them, we will leave the relations between forecasting questions $\mathcal{R}$ implicit by constructing the sentences directly. E,g, $\mathcal{R}(x_1, x_2) := x_1 = \neg x_2$ is implied by simply writing $x_1, x_2$ as $P, \neg P$.

*Table 2.* Consistency checks and the logical consistency conditions.

| Name | Tuple | Condition ($\mathcal{S}$) |
|---|---|---|
| NEGATION | $(P, \neg P)$ | $\mathbb{F}(P) + \mathbb{F}(\neg P) = 1$ |
| ANDOR | $(P, Q, P \wedge Q, P \vee Q)$ | $\mathbb{F}(P) + \mathbb{F}(Q) = \mathbb{F}(P \vee Q) + \mathbb{F}(P \wedge Q)$ |
| AND | $(P, Q, P \wedge Q)$ | $\max(\mathbb{F}(P) + \mathbb{F}(Q) - 1, 0) \leq \mathbb{F}(P \wedge Q) \leq \min(\mathbb{F}(P), \mathbb{F}(Q))$ |
| OR | $(P, Q, P \vee Q)$ | $\max(\mathbb{F}(P), \mathbb{F}(Q)) \leq \mathbb{F}(P \vee Q) \leq \min(1, \mathbb{F}(P) + \mathbb{F}(Q))$ |
| BUT | $(P, \neg P \wedge Q, P \vee Q)$ | $\mathbb{F}(P \vee Q) = \mathbb{F}(P) + \mathbb{F}(\neg P \wedge Q)$ |
| COND | $(P, Q\|P, P \wedge Q)$ | $\mathbb{F}(P)\mathbb{F}(Q\|P) = \mathbb{F}(P \wedge Q)$ |
| CONDCOND | $(P, Q\|P, R\|P \wedge Q)$ | $\mathbb{F}(P)\mathbb{F}(Q\|P)\mathbb{F}(R\|P \wedge Q) = \mathbb{F}(P \wedge Q \wedge R)$ |
| CONSEQUENCE $\mathcal{R}(P, Q) := P \implies Q$ | $(P, Q)$ | $\mathbb{F}(P) \leq \mathbb{F}(Q)$ |
| PARAPHRASE $\mathcal{R}(P, Q) := P \iff Q$ | $(P, Q)$ | $\mathbb{F}(P) = \mathbb{F}(Q)$ |

We do not include a specific consistency check for Bayesian updates, as we regard this as subsumed by COND.

## B. Arbitrage as a violation metric

We regard the forecaster's probabilities $\mathbb{F}(x_1), \mathbb{F}(x_2), \ldots$ as the spot prices offered on a prediction market with a proper scoring rule (taken to be $s(p) = \log(p)$ by default)[5] for $x_1, x_2$, etc. If these forecasts are inconsistent, then an arbitrageur can profit from bringing the market prices to consistency. In particular, there is a minimum guaranteed risk-free profit, called *arbitrage*, that the forecaster can be guaranteed to obtain regardless of the outcome of $x_1, x_2$ etc., i.e. without taking a position on the base outcomes. This minimum guaranteed arbitrage can be taken to be a consistency violation metric.

**Definition B.1** (Arbitrage-based Violation Metric). Let $\mathcal{R} : \text{Prop}^n \rightarrow \{\top, \bot\}$ be an n-ary relation such that $\mathcal{R}(\theta(x_1), \ldots \theta(x_n))$ is satisfied by the ground-truth resolutions $\theta : \text{Prop} \rightarrow \Theta$ for all tuples $(x_1, \ldots x_n)$. [6] Let $s : \text{Prop} \times \Theta \times [0,1] \rightarrow \mathbb{R}$ be a proper scoring rule that gives the score earned based on the probability assigned to the true resolution, e.g. $s(P, \theta, p(\theta)) = \log p(\theta)$. Let $(x_1, \ldots x_n) \in \text{Prop}^n$ be a question tuple, and denote $\Omega := \{\omega \in \Theta'^n \mid \mathcal{R}(\omega)\}$ the set of possible consistent resolutions (including None resolutions) of this tuple. Then for forecasts $(\mathbb{F}(x_1), \ldots \mathbb{F}(x_n))$ the arbitraged forecasts $(p_1 \ldots p_n)$ and the minimum guaranteed profit of the arbitrageur $\mathcal{V}(\mathbb{F}(x_1), \ldots \mathbb{F}(x_n))$ are given by:

$$(\arg\max, \max)_{p \in \text{Forecast}^n} \min_{\omega \in \Omega} \sum_{i=1}^{n} s(x_i, \omega_i, p_i(\omega_i)) - s(x_i, \omega_i, \mathbb{F}(x_i)(\omega_i)) \tag{1}$$

Where by convention, any score on a resolution $\omega_i = \text{None}$ is taken to be 0.

Definition B.1 is presented in full generality: $p$ and $\mathbb{F}(x_i)$ here are *probability distributions* on $\Theta$. Breaking it down: each $s(x_i, \omega_i, p_i(\omega_i)) - s(x_i, \omega_i, \mathbb{F}(x_i)(\omega_i))$ gives the arbitrageur's profit on the market for question $x_i$, given that it resolves $\omega_i$. The profit is summed across all markets in the tuple then minimized over all consistent worlds; this minimum is maximized across all possible arbitrageur bets.

It is helpful to explicitly state Eq 1 in the case of binary forecasting questions, as follows.

$$(\arg\max, \max)_{p \in [0,1]^n} \min_{\omega \in \Omega} \sum_{i=1}^{n} (s(p_i) - s(\mathbb{F}(x_i))) \delta_{\omega(i)=\top} + (s(1 - p_i) - s(1 - \mathbb{F}(x_i))) \delta_{\omega(i)=\bot} \tag{2}$$

We will illustrate our violation metric with two specific examples, for NEGATION and PARAPHRASE. For other consistency checks, the math becomes too convoluted and we use a numerical method in our project code.

### B.1. ParaphraseChecker

Let $P$ and $Q$ be equivalent sentences, and suppose that the forecaster produces forecasts $\mathbb{F}(P)$ and $\mathbb{F}(Q)$. A trader who instead brings prices to $\mathbb{F}'(P) = \mathbb{F}'(Q) = p$ for both questions earns a combined profit on both questions:

$$\begin{cases} s(p) - s(\mathbb{F}(P)) + s(p) - s(\mathbb{F}(Q)) & \text{if } P \\ s(1 - p) - s(1 - \mathbb{F}(P)) + s(1 - p) - s(1 - \mathbb{F}(Q)) & \text{if } \neg P \end{cases} \tag{3}$$

For this first example, we can graph this profit as a function of $p$ for illustration, shown in Fig. 1 – demonstrating that any $p \in (0.529, 0.576)$ is profitable for the arbitrageur, and further that the arbitrageur can *guarantee* a minimum profit of $0.095$ regardless of the outcome of $P$ by choosing the consistent probability $p = 0.555$.

We may compute this intersection analytically:

---

[5]A proper scoring rule, introduced in (Savage, 1971), is one that incentivizes honest reporting of probabilities. See (Hanson, 2002) for the logarithmic scoring rule that we use.

[6]This is well-defined because resolutions can be taken as a subset $\Theta \subseteq \text{Prop}$, by treating them as forecasting questions that always resolve to themselves by definition. For example, the forecasting question $\top$ is always worth \$1 and the forecasting question $\bot$ is always worth \$0.

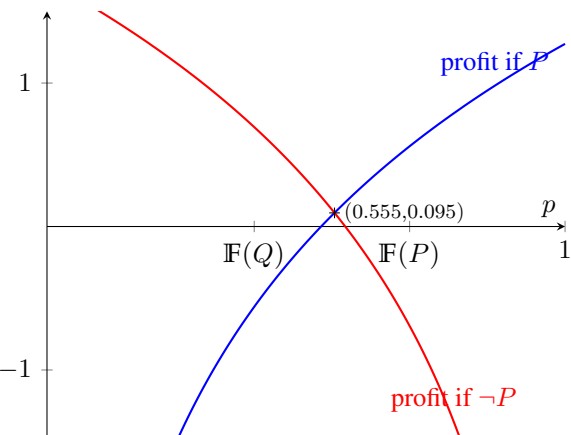

Figure 1. Profit earned by the arbitrageur in case of inconsistency over ParaphraseChecker, taking $s(p) = \log(p)$ and $\mathbb{F}(P), \mathbb{F}(Q) = 0.7, 0.4$ in (3).

.

$$s\left(p\right) - s\left(\mathbb{F}(P)\right) + s\left(p\right) - s\left(\mathbb{F}(Q)\right) = s\left(1-p\right) - s\left(1 - \mathbb{F}(P)\right) + s\left(1-p\right) - s\left(1 - \mathbb{F}(Q)\right)$$

$$2\log\frac{p}{1-p} = \log\frac{\mathbb{F}(P)\mathbb{F}(Q)}{(1 - \mathbb{F}(P))(1 - \mathbb{F}(Q))}$$

$$p = \frac{\sqrt{\mathbb{F}(P)\mathbb{F}(Q)}}{\sqrt{\mathbb{F}(P)\mathbb{F}(Q)} + \sqrt{(1 - \mathbb{F}(P))(1 - \mathbb{F}(Q))}}$$

Substituting this back into either expression in (3) we get the expression for the arbitrage:

$$\mathcal{V}(\mathbb{F}(P), \mathbb{F}(Q)) = -2\log\left(\sqrt{\mathbb{F}(P)\mathbb{F}(Q)} + \sqrt{(1 - \mathbb{F}(P))(1 - \mathbb{F}(Q))}\right) \tag{4}$$

**B.2. NegChecker**

Suppose the forecaster produces forecasts $\mathbb{F}(P)$ and $\mathbb{F}(\neg P)$. A trader who instead brings prices to $\mathbb{F}'(P) = p$, $\mathbb{F}'(\neg P) = 1 - p$ earns a combined profit on both questions:

$$\begin{cases} s\left(p\right) - s\left(\mathbb{F}(P)\right) + s\left(p\right) - s\left(1 - \mathbb{F}(\neg P)\right) & \text{if } P \\ s\left(1-p\right) - s\left(1 - \mathbb{F}(P)\right) + s\left(1-p\right) - s\left(\mathbb{F}(\neg P)\right) & \text{if } \neg P \end{cases} \tag{5}$$

Equating them and solving as before,

$$2\log\frac{p}{1-p} = \log\frac{\mathbb{F}(P)(1 - \mathbb{F}(\neg P))}{(1 - \mathbb{F}(P))\mathbb{F}(\neg P)}$$

$$p = \frac{\sqrt{\mathbb{F}(P)(1 - \mathbb{F}(\neg P))}}{\sqrt{\mathbb{F}(P)(1 - \mathbb{F}(\neg P))} + \sqrt{(1 - \mathbb{F}(P))\mathbb{F}(\neg P)}}$$

Substituting into (5), we get:

**Algorithm 1** Numerical computation of Eq 1

**procedure** ARBITRAGE($\theta, f, p$)
    ▷ Computes the profit earned by an arbitrageur who bets $p \in [0,1]^n$ against forecasts $f \in [0,1]^n$ upon an $n$-tuple of questions resolving as $\theta \in \Theta'^n$
    Initialize $s \leftarrow 0$
    **for** $i \in \{1 \ldots n\}$ **do**
        **if** $\theta(i) = \text{None}$ **then**
            continue
        **end if**
        **if** $\theta(i) = \top$ **then**
            $s \leftarrow s + s(p(i)) - s(f(i))$
        **end if**
        **if** $\theta(i) = \bot$ **then**
            $s \leftarrow s + s(1 - p(i)) - s(1 - f(i))$
        **end if**
    **end for**
    return $s$
**end procedure**
**procedure** MINARBITRAGE($f, p$)
    ▷ Minimizes ARBITRAGE($\theta, f, p$) over all $\theta$s that satisfy $\mathcal{R}$
    $\Omega := \{\omega \in \Theta'^n \mid \mathcal{R}(\omega)\}$               ▷ All $\Theta$ that satisfy the consistency check $\mathcal{R}$
    return $\min_{\theta \in \Omega}$ ARBITRAGE($\theta, f, p$)             ▷ Minimization over a finite set
**end procedure**
**procedure** MAXMINARBITRAGE($f$)
    ▷ Maximizes MINARBITRAGE($f, p$) over all possible arbitrageur bets $p$, i.e. calculates the arbitrageur's optimal bet and its profit thereof
    Initialize $p \leftarrow f$                       ▷ Initial guess
    $D \leftarrow [\epsilon, 1 - \epsilon]^n$                 ▷ Avoid $\log(0)$
    $p, v \leftarrow$ GLOBALMAXIMIZATION(MINARBITRAGE($f, p$))     ▷ maximize over $p$
    return $p, v$
**end procedure**

$$\mathcal{V}(\mathbb{F}(P), \mathbb{F}(\neg P)) = -2 \log \left( \sqrt{\mathbb{F}(P)(1 - \mathbb{F}(\neg P))} + \sqrt{(1 - \mathbb{F}(P))\mathbb{F}(\neg P)} \right) \tag{6}$$

### B.3. Numerical estimation

Explicitly deriving the violation metrics for other checkers from Eq 1 is infeasible by hand, and the expressions yielded by SymPy are very convoluted. We compute the violation numerically for all Checkers, i.e. use a global optimizer to compute the max. The global maximization algorithm we use is *shgo* (Simplicial Homology Global Optimization, introduced in (Endres et al., 2018)) as provided by SciPy; other methods we found to be equally effective were *differential_evolution* (Storn & Price, 1997) and *dual_annealing* (Xiang et al., 2013). The pseudocode for computing arbitrage, is given in Algorithm 1.

## C. Frequentist consistency metric

In a deterministic world, we cannot let any inconsistency pass; every time we prove any rule of probability does not hold exacly, we must discard the forecaster as flawed. This is too strict for the consistency check framework to be useful. Instead, we propose a violation metric and the corresponding inconsistency threshold based on statistical hypothesis testing.

Assume that each event $P$ has a true probability value $\mathbb{T}(P)$, say under some world model that accounts for aleatoric uncertainty.

**Definition C.1** (Frequentist consistency). A frequentist-consistent forecaster $\mathbb{F}$ samples a Gaussian estimate $\mathbb{T}(P) + \varepsilon$ of each event $P$, with variance $\sigma^2 \mathbb{T}(P)(1 - \mathbb{T}(P))$ for a hyperparameter $\sigma^2$:

$$\mathbb{F}(P) - \mathbb{T}(P) \sim \mathsf{N}\left(0, \sigma^2 \mathbb{T}(P)(1 - \mathbb{T}(P))\right) \quad \text{independently for all events } P. \tag{7}$$

This is principled from the frequentist perspective. Consider a forecaster that just samples the (relevant subset of) the world $n$ times using the best available world simulator, and estimates the probability of each event $P$ as the proportion of times that $P$ occurs in the $n$ samples. If we estimate the probability as the average chance of an event $P$ with true probability $p$ occurring out of $n$ times, then this estimate has a scaled binomial distribution with mean $p$ and variance $p(1-p)/n$. To reach Equation (7), replace the averaged binomial with the Gaussian of the same variance, and denote $\sigma^2 := 1/n$.

This simple model enables us to derive hypothesis tests for each of the consistency checks described in 2. The null hypothesis is always that the forecaster is frequentist-consistent. Note that $\sigma^2$ is not our estimate of the variance of any forecaster; it is just a hyperparameter that controls how strict our null hypothesis is. We leave estimating the variance of a particular forecaster and testing frequentist consistency based on that alone to future work.

**Notation** The expression $a\mathsf{N}(0, c^2)$ denotes a Gaussian random variable with mean 0 and variance $a^2 c^2$. The expression $a\mathsf{N}(0, c^2) + b\mathsf{N}(0, c^2)$ denotes a Gaussian random variable with mean 0 and variance $a^2 c^2 + b^2 c^2$. All sums range over the cyclic permutations of the variables under the sum. All $\mathsf{N}(0, c^2)$ terms appearing with the same power of $\sigma$ are independent. Two $\mathsf{N}(0, c^2)$ terms appearing with a different power of $\sigma$ may be correlated; this is not important for our purposes, since we discard high-order powers of $\sigma$.

**Bootstrapping the true probability** The final expressions for hypothesis test statistics might involve the true probability $\mathbb{T}(P)$. It is not available, so we just plug in $\mathbb{F}(P)$ for $\mathbb{T}(P)$ in the end. If we had a prior on $\mathbb{T}(P)$, we could combine it with $\mathbb{F}(P)$ to get a more robust estimate.

**NEGATION** We take the violation metric and the corresponding threshold as to produce a hypothesis test against this:

$$\mathbb{F}(P) + \mathbb{F}(\neg P) - 1 = \mathbb{T}(P) + \varepsilon_1 + \mathbb{T}(\neg P) + \varepsilon_2 - 1 = \varepsilon_1 + \varepsilon_2 \sim \mathsf{N}\left(0, \sigma^2(\mathbb{T}(P)(1 - \mathbb{T}(P)) + \mathbb{T}(\neg P)(1 - \mathbb{T}(\neg P)))\right)$$

We estimate the unknown $\mathbb{T}$ values with the corresponding $\mathbb{F}$ estimates. Note that, although $\mathbb{T}(P) = 1 - \mathbb{T}(\neg P)$, it is of course not necessarily the case that $\mathbb{F}(P) = 1 - \mathbb{F}(\neg P)$.

The error distribution is $\sigma\mathsf{N}\left(\mathbb{F}(P)(1 - \mathbb{F}(P)) + \mathbb{F}(\neg P)(1 - \mathbb{F}(\neg P))\right)$, and the two-sided test is

$$|\mathbb{F}(P) + \mathbb{F}(\neg P) - 1| < \gamma\sigma\sqrt{(1 - \mathbb{F}(P))\mathbb{F}(P) + (1 - \mathbb{F}(\neg P))\mathbb{F}(\neg P)}$$

for some scale factor $\gamma$ (number of standard deviations) that scales the power of the test. For example, $\gamma = 3$ gives a 99.7%-confidence interval.

We now want to compute some *consistency violation metric* that makes inconsistency comparable across different checks. The natural idea is to aggregate all terms dependent on $\mathbb{F}$ to one side; and make the hypothesis test just a threshold on the computed violation metric.

It is possible that the denominator of the resulting expression is 0 when the forecaster is certain and $\mathbb{F}$ is 0 or 1; to avoid division with zero, we add a small regularization term $\beta_{\text{MIN}} = 10^{-3}$. For the discussion of hyperparameters, see the last paragraph of this section.

Our consistency violation metric is then:

$$v_{\text{NEGATION}} = \frac{|\mathbb{F}(P) + \mathbb{F}(\neg P) - 1|}{\sqrt{(1 - \mathbb{F}(P))\mathbb{F}(P) + (1 - \mathbb{F}(\neg P))\mathbb{F}(\neg P) + \beta_{\text{MIN}}}}.$$

The hyperparameter $\sigma^2$ determines how strict we are with rejecting inconsistencies which could be attributed to "noisy" predictions. Note that the violation metric itself does not depend on $\sigma^2$.

A violation (inconsistency), therefore, occurs when:

$$v_{\text{NEGATION}} > \gamma\sigma.$$

**CONDCOND**    This is a more complex consistency check; we derive the hypothesis test and violation metric in detail below. For the other checks, we just report the short derivation.

$$(a, b, c, d) = (\mathbb{T}(P), \mathbb{T}(Q \mid P), \mathbb{T}(R \mid P \wedge Q), \mathbb{T}(P \wedge Q \wedge R))$$
$$(a', b', c', d') = (\mathbb{F}(P), \mathbb{F}(Q \mid P), \mathbb{F}(R \mid P \wedge Q), \mathbb{F}(P \wedge Q \wedge R))$$

We can write:

$$\mathbb{F}(P) = \mathsf{N}\left(0, \sigma^2 a(1 - a)\right) + a,$$
$$\mathbb{F}(Q \mid P) = \mathsf{N}\left(0, \sigma^2 b(1 - b)\right) + b,$$
$$\mathbb{F}(R \mid P \wedge Q) = \mathsf{N}\left(0, \sigma^2 c(1 - c)\right) + c,$$
$$\mathbb{F}(P \wedge Q \wedge R) = \mathsf{N}\left(0, \sigma^2 d(1 - d)\right) + d$$

We now compute the difference of the two expressions that should be equal. All sums and products are cyclic over $a$, $b$, $c$.

$$\mathbb{F}(P)\mathbb{F}(Q \mid P)\mathbb{F}(R \mid P \wedge Q) - \mathbb{F}(P \wedge Q \wedge R) = abc - d$$
$$+ \sigma\left(\sum_a bc\mathsf{N}(0, a(1 - a)) - \mathsf{N}(0, d(1 - d))\right)$$
$$+ \sigma^2 \sum_a \mathsf{N}(0, b(1 - b))\mathsf{N}(0, c(1 - c))$$
$$+ \sigma^3 \prod_a \mathsf{N}(0, a(1 - a)).$$

In the above, all Gaussians with the same variance are identical, and all other combinations are independent. As $abc - d = 0$ by the law of total probability, the leading error term is next to $\sigma$. This is a Gaussian with mean 0 and standard deviation:

$$\sigma\sqrt{\sum_a bca(1 - a) + d(1 - d)}.$$

We now discard the terms of $\sigma^2$, $\sigma^3$, and in general any higher order power of $\sigma$. This is principled because the coefficients can always be (in some confidence interval) upper bounded by a constant independent of $\sigma$. Hence, if $\sigma$ is small enough, the resulting test will be very close to the true hypothesis test. Note with properties of Gaussian distributions, even though we subtract errors, we add variances.

We do not have the true probabilities $a$, $b$, $c$, $d$, so we just plug in $(a', b', c', d') = (\mathbb{F}(P), \mathbb{F}(Q \mid P), \mathbb{F}(R \mid P \wedge Q), \mathbb{F}(P \wedge Q \wedge R))$. Thus the hypothesis test is (where the sum is cyclic over $a'$, $b'$, $c'$):

$$|a'b'c' - d'| > \gamma\sigma\sqrt{\sum_{a'} b'c'a'(1-a') + d'(1-d')}$$

Our violation metric is then:

$$v_{\text{CONDCOND}} = \frac{|a'b'c' - d'|}{\sqrt{\sum_{a'} b'c'a'(1-a') + d'(1-d') + \beta_{\text{MIN}}}}.$$

where again $(a', b', c', d') = (\mathbb{F}(P), \mathbb{F}(Q \mid P), \mathbb{F}(R \mid P \wedge Q), \mathbb{F}(P \wedge Q \wedge R))$ are the forecasts.

**COND** Similarly as for CONDCOND: we denote $(a, b, c) = (\mathbb{T}(P), \mathbb{T}(P \mid Q), \mathbb{T}(P \wedge Q))$ and the associated $(a', b', c')$ for the forecasts. Then we can compute

$$\mathbb{F}(P)\mathbb{F}(Q \mid P) - \mathbb{F}(P \wedge Q)$$
$$= ab - c + \sigma\left(b\mathsf{N}(0, a(1-a) + a\mathsf{N}(0, b(1-b)) - \mathsf{N}(0, c(1-c))\right) + \sigma^2\mathsf{N}(0, a(1-a))\mathsf{N}(0, b(1-b)).$$

The term next to $\sigma$ is a Gaussian with mean 0 and standard deviation:

$$\sigma\sqrt{b(1-b) + a(1-a) + c(1-c)}.$$

Again, we have to plug in $(a', b', c') = (\mathbb{F}(P), \mathbb{F}(Q \mid P), \mathbb{F}(P \wedge Q))$ instead of $(a, b, c)$.

Our violation metric is then:

$$v_{\text{COND}} = \frac{|a'b' - c'|}{\sqrt{b'(1-b') + a'(1-a') + c'(1-c') + \beta_{\text{MIN}}}}$$

And the test is again, for a suitable $\gamma$ corresponding to the desired power of the test:

$$v_{\text{COND}} > \gamma\sigma.$$

**PARAPHRASE** Here we can simply check whether $P$ and $Q$ are the same.

$$\mathbb{F}(P) - \mathbb{F}(Q) = \mathbb{T}(P) + \varepsilon_1 - \mathbb{T}(Q) - \varepsilon_2 = \varepsilon_1 - \varepsilon_2 \sim \mathsf{N}\left(0, \sigma^2((\mathbb{T}(P)(1 - \mathbb{T}(P)) - (\mathbb{T}(Q)(1 - \mathbb{T}(Q)))\right)$$

This yields the violation metric:

$$v_{\text{PARAPHRASE}} = \frac{|\mathbb{F}(P) - \mathbb{F}(Q)|}{\sqrt{(\mathbb{F}(P)(1 - \mathbb{F}(P)) + (\mathbb{F}(Q)(1 - \mathbb{F}(Q)) + \beta_{\text{MIN}}}}.$$

**ANDOR**

$$\mathbb{F}(P) + \mathbb{F}(Q) - \mathbb{F}(P \vee Q) - \mathbb{F}(P \wedge Q) = \mathbb{T}(P) + \mathbb{T}(Q) - \mathbb{T}(P \vee Q) - \mathbb{T}(P \wedge Q) + \varepsilon_1 + \varepsilon_2 - \varepsilon_3 - \varepsilon_4$$
$$= \varepsilon_1 + \varepsilon_2 - \varepsilon_3 - \varepsilon_4$$
$$\sim \mathsf{N}\left(0, \sigma^2((\mathbb{T}(P)(1 - \mathbb{T}(P)) + (\mathbb{T}(Q)(1 - \mathbb{T}(Q) - (\mathbb{T}(P \vee Q)(1 - \mathbb{T}(P \vee Q)) - (\mathbb{T}(P \wedge Q)(1 - \mathbb{T}(P \wedge Q)))\right).$$

We again plug in $\mathbb{F}$ instead of $\mathbb{T}$ to compute the error term allowed:

$$\gamma\sigma\sqrt{\mathbb{F}(P)(1-\mathbb{F}(P))+\mathbb{F}(Q)(1-\mathbb{F}(Q)+\mathbb{F}(P\vee Q)(1-\mathbb{F}(P\vee Q))+\mathbb{F}(P\wedge Q)(1-\mathbb{F}(P\wedge Q))}$$

and violation metric:

$$v_{\text{ANDOR}}=\frac{|\mathbb{F}(P)+\mathbb{F}(Q)-\mathbb{F}(P\vee Q)-\mathbb{F}(P\wedge Q)|}{\sqrt{\mathbb{F}(P)(1-\mathbb{F}(P))+\mathbb{F}(Q)(1-\mathbb{F}(Q)+\mathbb{F}(P\vee Q)(1-\mathbb{F}(P\vee Q)+\mathbb{F}(P\wedge Q)(1-\mathbb{F}(P\wedge Q)+\beta_{\text{MIN}}}}.$$

**BUT**

$$\mathbb{F}(P\vee Q)-\mathbb{F}(P)-\mathbb{F}(\neg P\wedge Q)=\mathbb{T}(P\vee Q)-\mathbb{T}(P)-\mathbb{T}(\neg P\wedge Q)+\varepsilon_1-\varepsilon_2-\varepsilon_3=$$
$$\varepsilon_1-\varepsilon_2-\varepsilon_3\sim$$
$$\mathsf{N}\left(0,\sigma^2((\mathbb{T}(P\vee Q)(1-\mathbb{T}(P\vee Q))-(\mathbb{T}(P)(1-\mathbb{T}(P))-(\mathbb{T}(\neg P\wedge Q)(1-\mathbb{T}(\neg P\wedge Q)))\right)$$

with error term:

$$\gamma\sigma\sqrt{\mathbb{F}(P\vee Q)(1-\mathbb{F}(P\vee Q)+\mathbb{F}(P)(1-\mathbb{F}(P)+\mathbb{F}(\neg P\wedge Q)(1-\mathbb{F}(\neg P\wedge Q)}$$

and violation metric:

$$v_{\text{BUT}}=\frac{|\mathbb{F}(P\vee Q)-\mathbb{F}(P)-\mathbb{F}(\neg P\wedge Q)|}{\sqrt{\mathbb{F}(P\vee Q)(1-\mathbb{F}(P\vee Q))+\mathbb{F}(P)(1-\mathbb{F}(P))+\mathbb{F}(\neg P\wedge Q)(1-\mathbb{F}(\neg P\wedge Q)+\beta_{\text{MIN}}}}$$

**CONSEQUENCE**   In the case of inequalities involving $\leq$, there are two ways in which the consistency check can be passed. If $\mathbb{F}(P)\leq\mathbb{F}(Q)$, the consistency check is automatically passed. Otherwise, we check for pseudo-equality using the same violation metric as in PARAPHRASE.

$$v_{\text{CONSEQUENCE}}=[\mathbb{F}(P)>\mathbb{F}(Q)]\frac{|\mathbb{F}(P)-\mathbb{F}(Q)|}{\sqrt{\mathbb{F}(P)(1-\mathbb{F}(P))+\mathbb{F}(Q)(1-\mathbb{F}(Q))+\beta_{\text{MIN}}}}$$

where $[\mathbb{F}(P)>\mathbb{F}(Q)]$ is the Iverson Bracket (1 if true, 0 otherwise).

**AND**   Similarly to CONSEQUENCE, if the chain of strict inequalities

$$\max(\mathbb{F}(P)+\mathbb{F}(Q)-1,0)<\mathbb{F}(P\wedge Q)<\min(\mathbb{F}(P),\mathbb{F}(Q))$$

holds, then the check automatically passes. We set $v_{\text{AND\_LHS}}=0$ and $v_{\text{AND\_RHS}}=0$ if it passes the first and second strict inequality respectively.

If not, then we test for pseudo-equality for the violating pair:

LHS : $\max(\mathbb{F}(P)+\mathbb{F}(Q)-1,0)=\mathbb{F}(P\wedge Q)$

RHS : $\mathbb{F}(P\wedge Q)=\min(\mathbb{F}(P),\mathbb{F}(Q))$

Equality check if it fails the first inequality:

$$\varepsilon_{\text{LHS}}=\begin{cases}\gamma\sigma\sqrt{\mathbb{F}(P)(1-\mathbb{F}(P))+\mathbb{F}(Q)(1-\mathbb{F}(Q))+\mathbb{F}(P\wedge Q)(1-\mathbb{F}(P\wedge Q))}, & \text{if }\mathbb{F}(P)+\mathbb{F}(Q)-1>0,\\ \text{N/A}, & \text{otherwise pass as }\mathbb{F}(P\wedge Q)\geq 0.\end{cases}$$

$$v_{\text{AND\_LHS}} = [\mathbb{F}(P) + \mathbb{F}(Q) - 1 > \mathbb{F}(P \wedge Q)] \frac{\mathbb{F}(P) + \mathbb{F}(Q) - 1 - \mathbb{F}(P \wedge Q)}{\sqrt{\mathbb{F}(P)(1 - \mathbb{F}(P)) + \mathbb{F}(Q)(1 - \mathbb{F}(Q)) + \mathbb{F}(P \wedge Q)(1 - \mathbb{F}(P \wedge Q)) + \beta_{\text{MIN}}}}$$

Equality check if it fails the second inequality:

Define $\mathbb{F}(R) = \min(\mathbb{F}(P), \mathbb{F}(Q))$.

$$\varepsilon_{\text{RHS}} = \gamma\sigma\sqrt{\mathbb{F}(P \wedge Q)(1 - \mathbb{F}(P \wedge Q)) + \mathbb{F}(R)(1 + \mathbb{F}(R))}$$

$$v_{\text{AND\_RHS}} = [\mathbb{F}(R) < \mathbb{F}(P \wedge Q)] \frac{\mathbb{F}(P \wedge Q) - \mathbb{F}(R)}{\sqrt{\mathbb{F}(P \wedge Q)(1 - \mathbb{F}(P \wedge Q)) + \mathbb{F}(R)(1 - \mathbb{F}(R)) + \beta_{\text{MIN}}}}$$

Consistency is violated if either inequality is violated, *and* the respective hypothesis test for pseudo-equality fails. We use $v_{\text{AND\_LHS}}$ for the first and $v_{\text{AND\_RHS}}$ for the second inequality. We define $v_{\text{AND}} = \max\{v_{\text{AND\_LHS}}, v_{\text{AND\_RHS}}\}$.

**OR**  We proceed similarly as for AND.

If the strict inequality $\max(\mathbb{F}(P), \mathbb{F}(Q)) < \mathbb{F}(P \vee Q) < \min(1, \mathbb{F}(P) + \mathbb{F}(Q))$ holds, then it automatically passes. We set $v_{\text{OR\_LHS}} = 0$ and $v_{\text{OR\_RHS}} = 0$ if it passes the first and second strict inequality respectively.

If not, we test for pseudo-equality:

LHS : $\max(\mathbb{F}(P), \mathbb{F}(Q)) = \mathbb{F}(P \vee Q)$

RHS : $\mathbb{F}(P \vee Q) = \min(1, \mathbb{F}(P) + \mathbb{F}(Q))$.

Equality check LHS: Define $\mathbb{F}(S) = \max(\mathbb{F}(P), \mathbb{F}(Q))$.

$$\varepsilon_{\text{LHS}} = \gamma\sigma\sqrt{\mathbb{F}(S)(1 - \mathbb{F}(S)) + \mathbb{F}(P \vee Q)(1 - \mathbb{F}(P \vee Q))}$$

$$v_{\text{OR\_LHS}} = [\mathbb{F}(S) > \mathbb{F}(P \vee Q)] \frac{\mathbb{F}(S) - \mathbb{F}(P \vee Q)}{\sqrt{\mathbb{F}(S)(1 - \mathbb{F}(S)) + \mathbb{F}(P \vee Q)(1 - \mathbb{F}(P \vee Q)) + \beta_{\text{MIN}}}}$$

Equality check RHS:

$$\varepsilon_{\text{RHS}} = \begin{cases} \gamma\sigma\sqrt{\mathbb{F}(P \vee Q)(1 - \mathbb{F}(P \vee Q)) + \mathbb{F}(P)(1 - \mathbb{F}(P)) + \mathbb{F}(Q)(1 - \mathbb{F}(Q))}, & \text{if } \mathbb{F}(P) + \mathbb{F}(Q) < 1, \\ \text{N/A}, & \text{otherwise pass as } \mathbb{F}(P \vee Q) \leq 1. \end{cases}$$

$$v_{\text{OR\_RHS}} = [\mathbb{F}(P) + \mathbb{F}(Q) < \mathbb{F}(P \vee Q)] \frac{\mathbb{F}(P \vee Q) - \mathbb{F}(P) - \mathbb{F}(Q)}{\sqrt{\mathbb{F}(P \vee Q)(1 - \mathbb{F}(P \vee Q)) + \mathbb{F}(P)(1 - \mathbb{F}(P)) + \mathbb{F}(Q)(1 - \mathbb{F}(Q)) + \beta_{\text{MIN}}}}$$

Consistency is violated if either inequality is violated, *and* the subsequent hypothesis test for pseudo-equality fails. We use $v_{\text{OR\_LHS}}$ for the first and $v_{\text{OR\_RHS}}$ for the second inequality. Analogously to AND, define $v_{\text{OR}} = \max\{v_{\text{OR\_LHS}}, v_{\text{OR\_RHS}}\}$.

**Hyperparameters for hypothesis testing**  Our goal is for the rejection criteria to be similar to the arbitrage violation metric in Appendix B on simple examples. We choose $\gamma = 2.58$ for all checks, to ensure 99%-confidence intervals for two-sided tests; future work may consider using a different $\gamma$ for checks that require one-sided tests. We pick $\sigma = 0.05$ (corresponding to $n = 400$ in Definition C.1). The allowed violation threshold for all checks is then $\gamma\sigma = 0.129$. For reference, a NEGATION pair $(\mathbb{F}(P), \mathbb{F}(\neg P)) = (0.5, 0.59)$ has a violation metric of 0.128, and would thus not be rejected as inconsistent. This exactly corresponds to the tolerance threshold of $10^{-2}$ of profit for the arbitrage metric, described in Section 2.

We pick $\beta_{\text{MIN}} = 10^{-3}$ because LLM forecasters from Halawi et al. (2024) answer with at most 3 digits of precision for events close to 0 and 1 in probability.

## D. Results tables

This section contains our results for all forecasters we tested, besides the advanced set-up in (Halawi et al., 2024), i.e. without retrieval-augmented generation nor chain-of-thought. The forecaster simply gets a forecasting question (title, body, resolution date), and needs to output a single number between 0 and 1. We use the Instructor library (Liu et al., 2024) and JSON API calls to enforce correctly formatted outputs.

*Table 3.* Advanced Forecaster set-up from (Halawi et al., 2024)

| Checker | $n$ | Arbitrage metric | | | Frequentist metric | | |
| | | # violations | Violation (mean) | Violation (median) | # violations | Violation (mean) | Violation (median) |
|---|---|---|---|---|---|---|---|
| NEGATION | 80 | 42 | 0.0811 | 0.0120 | 43 | 0.2684 | 0.1550 |
| ANDOR | 80 | 47 | 0.0572 | 0.0139 | 52 | 0.2496 | 0.1741 |
| BUT | 80 | 61 | 0.0823 | 0.0441 | 64 | 0.3404 | 0.2939 |
| AND | 80 | 10 | 0.0130 | 0.0000 | 12 | 0.0582 | 0.0000 |
| OR | 80 | 26 | 0.0428 | 0.0012 | 28 | 0.1563 | 0.0708 |
| COND | 80 | 35 | 0.0288 | 0.0063 | 31 | 0.1252 | 0.0809 |
| CONDCOND | 80 | 25 | 0.0150 | 0.0000 | 32 | 0.1251 | 0.0836 |
| CONSEQUENCE | 80 | 4 | 0.0479 | 0.0000 | 4 | 0.0873 | 0.0000 |
| PARAPHRASE | 80 | 30 | 0.0100 | 0.0044 | 32 | 0.1032 | 0.0931 |

*Table 4.* Basic Forecaster (GPT-4o)

| Checker | $n$ | Arbitrage metric | | | Frequentist metric | | |
| | | # violations | Violation (mean) | Violation (median) | # violations | Violation (mean) | Violation (median) |
|---|---|---|---|---|---|---|---|
| NEGATION | 80 | 48 | 0.0371 | 0.0120 | 48 | 0.1968 | 0.1550 |
| ANDOR | 80 | 48 | 0.0472 | 0.0162 | 55 | 0.2450 | 0.1954 |
| BUT | 80 | 44 | 0.0445 | 0.0188 | 55 | 0.2467 | 0.2071 |
| AND | 80 | 2 | 0.0075 | 0.0000 | 2 | 0.0203 | 0.0000 |
| OR | 80 | 39 | 0.0364 | 0.0078 | 39 | 0.1760 | 0.1070 |
| COND | 80 | 28 | 0.0186 | 0.0057 | 13 | 0.0856 | 0.0671 |
| CONDCOND | 80 | 28 | 0.0132 | 0.0022 | 34 | 0.1314 | 0.1138 |
| CONSEQUENCE | 80 | 6 | 0.0234 | 0.0000 | 6 | 0.0556 | 0.0000 |
| PARAPHRASE | 80 | 27 | 0.0157 | 0.0000 | 28 | 0.1009 | 0.0000 |

*Table 5.* Basic Forecaster (GPT-3.5)

| Checker | $n$ | Arbitrage metric | | | Frequentist metric | | |
|---|---|---|---|---|---|---|---|
| | | # violations | Violation (mean) | Violation (median) | # violations | Violation (mean) | Violation (median) |
| NEGATION | 80 | 34 | 0.1940 | 0.0093 | 42 | 0.4476 | 0.1344 |
| ANDOR | 80 | 39 | 0.1271 | 0.0076 | 50 | 0.3923 | 0.2068 |
| BUT | 80 | 63 | 0.3092 | 0.0892 | 67 | 0.6501 | 0.4651 |
| AND | 80 | 8 | 0.0221 | 0.0000 | 13 | 0.0661 | 0.0000 |
| OR | 80 | 45 | 0.2013 | 0.0298 | 48 | 0.4938 | 0.2384 |
| COND | 80 | 26 | 0.0478 | 0.0037 | 21 | 0.1321 | 0.0762 |
| CONDCOND | 80 | 29 | 0.0224 | 0.0000 | 36 | 0.1649 | 0.0958 |
| CONSEQUENCE | 80 | 9 | 0.0211 | 0.0000 | 11 | 0.0660 | 0.0000 |
| PARAPHRASE | 80 | 28 | 0.0709 | 0.0000 | 32 | 0.1996 | 0.0000 |

# E. Data and prompt examples

### E.1. Data types

Figure 2 shows the data stored on forecasting questions. Of these, only *title* and *body* are shown to the forecaster.

---

**Forecasting question data type.**

- **id**: Universally Unique Question Identifier (UUID)

- **title**: Title of forecasting question.

- **body**: Generally resolution criterion, background information etc.

- **resolution_date**: Resolution date.

- **question_type**: Question type, indicating the type of forecasts. Exclusively *binary* or *conditional-binary* in this paper; other potential options: *multiple-choice*, *integer*, *continuous-value*, *opinion*.

- **data_source**: Source of question: either website from which it was scraped or *synthetic*.

- **url**: URL if question was scraped, else *null*

- **metadata**: Any additional flags, e.g. *topics* for Metaculus questions; *tags* and *category* for synthetic questions.

- **resolution**: Boolean (if question is already resolved), else *null*.

---

*Figure 2.* Description of the forecasting question data type.

| Example forecasting question (scraped) |
| --- |

- **id**: 07b11b15-6872-4280-a94f-17b6d15a1b8a

- **title**: Will SpaceX land people on Mars before 2030?

- **body**: This question will resolve as Yes if SpaceX successfully lands at least one human on the surface of Mars on or before December 31, 2030. The landing must be confirmed by SpaceX through an official announcement or live broadcast. The human(s) must be alive upon landing and must perform at least one extravehicular activity (EVA) on the Martian surface, which must be documented and released to the public. In the event of a dispute regarding the success of the mission, the resolution will defer to the judgment of an international space agency such as NASA or ESA. If no landing attempt is made by the specified date, or if all attempts fail to meet the above criteria, the question will resolve as No.

- **resolution_date**: 2030-12-31 23:59:59+00:00

- **question_type**: binary

- **data_source**: metaculus

- **url**: https://www.metaculus.com/questions/349

- **metadata**:
  - **topics**:
    * **id**: 184, **slug**: elon-musk, **name**: Elon Musk, **link_id**: 27681, **num_questions**: 159
    * **id**: 485, **slug**: spacex-reusable-launch-system-development-program, **name**: SpaceX reusable launch system, **link_id**: 27682, **num_questions**: 130
    * **id**: 1365, **slug**: spacex, **name**: SpaceX, **link_id**: 75197, **num_questions**: 112
    * **id**: 564, **slug**: colonization-of-mars, **name**: Colonization of Mars, **link_id**: 27683, **num_questions**: 70
    * **id**: 1768, **slug**: spacex-mars-transportation-infrastructure, **name**: SpaceX Mars transportation infrastructure, **link_id**: 40982, **num_questions**: 5

- **resolution**: null

*Figure 3.* Example of a scraped forecasting question data type.

> **Example forecasting question (synthetic)**

- **id**: 4b98368c-6287-47e0-8f9e-5917e2a24a3d

- **title**: Will Russia launch a manned mission to the Moon before 2030?

- **body**: This question will resolve as Yes if, before January 1, 2030, the Russian Federation successfully launches and completes a manned mission to the Moon, where 'successful' is defined as a mission where astronauts land on the lunar surface and return safely to Earth. The mission must be officially recognized by Roscosmos or another authoritative space agency. In the event of a joint mission involving Russia and other countries, the mission will still resolve as Yes if Russian astronauts are part of the crew that lands on the Moon. If no such mission is launched, or if a mission is launched but does not meet the above criteria, the question will resolve as No. In the case of ambiguity or lack of clear public information by the resolution date, the question will resolve as No unless official statements or evidence are provided by Roscosmos or an equivalent authoritative body that confirm the mission's success as per the defined criteria.

- **resolution_date**: 2030-12-31 23:59:59+00:00

- **question_type**: binary

- **data_source**: synthetic

- **url**: null

- **metadata**:
    - **tags**:
        * Russia
    - **categories**:
        * Space

- **resolution**: null

*Figure 4.* Example of a synthetic forecasting question. All question generations are seeded with the *metadata* field.

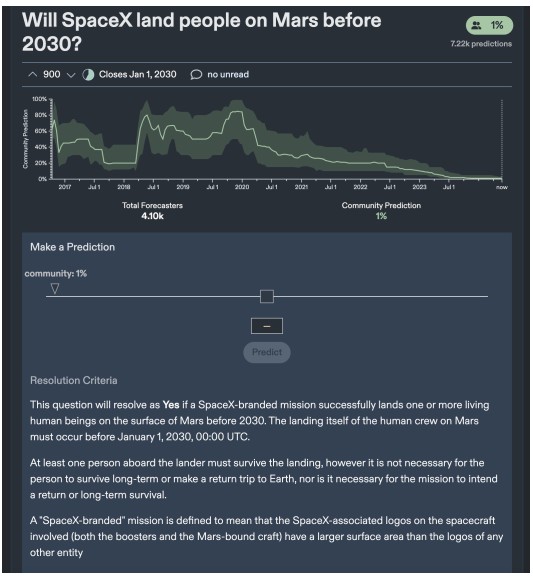

*Figure 5.* Example of a question from Metaculus

Example tuple (COND)

- **P**:
    - **id**: 9b40ebcd-71a8-4cfe-802c-578b946fcc66
    - **title**: Will North Korea engage in a significant diplomatic negotiation with a Western country by 2025?
    - ...

- **Q_given_P**:
    - **id**: 6da2b933-9183-46d4-a8da-5e10d6820b9e
    - **title**: Given North Korea engages in significant diplomatic negotiations with a Western country by 2025, will it significantly reform its legal system by 2030?
    - ...

- **P_and_Q**:
    - **id**: fafbaa5f-4ece-4526-a94d-fecb1b56f315
    - **title**: Will North Korea engage in significant diplomatic negotiations with a Western country by 2025 and significantly reform its legal system by 2030?
    - ...

*Figure 6.* Example of an instantiated CONDCOND forecasting question tuple.

### E.2. Prompts

In this section, we present the prompts used for the different parts of our pipeline. For each LLM call, we use the Instructor library (Liu et al., 2024) and JSON API calls to enforce correctly formatted Pydantic output objects. The whitespace in the figures is not representative of the whitespace in actual queries.

---

**Synthetic question generation prompt**

I want you to help me generate some forecasting questions for a forecasting market site like Metaculus or PredictIt. I will provide you with a category and some tags. Your task is to generate questions that can be answered with a probability between 0 and 1. For each tag, generate a relevant question if the tag is pertinent to the category. If the tag is not relevant, generate a general question about the category.
Examples:
{example_1}
{example_2}
{example_3}
{example_4}
{example_5}
{example_6}
Category: {category} Tags: {tags}

---

*Figure 7.* The prompt used for generating the *title* field of forecasting questions, given the *category* and *tags* metadata.

A list of initial quality-filtered questions is supplied to seed the list of examples.

---

**Tuple instantiation prompt – OR**

You are a helpful assistant. I will give you two forecasting questions with Yes/No answers. You should then give me the logical OR of these two questions, i.e. the question that would be answered YES if EITHER question is answered YES, and NO otherwise. Notes:

- Your response should be as clear as possible, since the words 'and' and 'or' are used ambiguously in natural language. For example, 'Will P happen or will Q happen? is usually confusing, as it sounds like you are asking which of the two will happen (whereas you're actually seeking a YES/NO answer on whether either of the two will happen). Instead, if there is any chance of confusion, you should give me something like: Will either of the following occur: (a) P (b) Q?

- When the questions allow for a simple rephrasing or factorization (e.g. using words like 'respectively', 'both' or 'either'), go for it.

- If one or both of the given questions is already a logical combination of questions, join them in the most natural way possible. E.g.

    - combine ((P1 OR P2) OR Q) how you would combine (P1 OR P2 OR Q)
    - ((P1 AND P2) OR Q) might have to be combined as something like: Will EITHER of the following occur: (1) BOTH of the following occur: (a) P1 AND (b) P2 (2) Q. Unless a more natural formulation exists.

- Be careful when combining conditional expressions (which often have words like 'given' and 'if'). '(Given A then P) OR (Given B then Q) should be combined as is, rather than messing up the conditions. E.g. a phrasing like 'Will either of the following occur: (a) Given A then P? (b) Given B then Q?' is good.

- This also applies when only one of the questions is conditional. Like 'P OR (Given A then Q)'should be phrased as something like: 'Will either of the following occur given their respective conditions are met? (a) P (b) Given A, then Q?'.

- Most importantly: make sure you retain ALL the information in the question bodies from BOTH base questions! You cannot discard a single relevant detail. All this is for an experiment to test the logical consistency of forecasters: The combined question you give will be handed to the forecasters without having seen the base questions, so it is critical that all the information in the base questions be included in your logical combination; the resolution criterion for each component should be neatly and clearly provided.

- Also, make sure that the title is self-sufficient independent of the body, i.e. is a question that can be meaningfully answered without looking at the body. So you CANNOT give me a question title like 'Is the following true?' or 'What will happen if the following happens?'

- One type of question you may be given is a single choice from a multiple choice question. For example, you may be given 'Which of these countries will legalize human cloning by 2030? (Japan)'. This is asking if Japan will recognize and legalize human cloning by 2030. Such a question may also itself be a logical combination – e.g. 'Which of these countries will legalize human cloning by 2030? (UK, France, or Germany) is asking if any either of the UK, France, or Germany will legalize human cloning by 2030. Make sure to correctly combine such combinations as previously described.

---

*Figure 8.* The prompt used for instantiating OR tuples. We use similar prompts for other checks.

> **Relevance scoring prompt**
>
> I'm doing a project that involve eliciting probabilities from LLMs to measure the calibration, consistency and such properties of LLM forecasters. As part of this project we will be taking logical combinations of forecasting questions and eliciting probabilities on them. I need your help in deciding, for two given forecasting questions, whether it makes sense to think about their logical combinations/whether it's worth doing so.
> For example, we might want to elicit the probability of
> 'Will Donald Trump win the 2024 US presidential election? AND Will US economic growth exceed 3.5% in 2025?'
> because Trump winning the election might potentially (positively or negatively) affect economic growth in the following year.
> But we probably wouldn't care about the probability of
> 'Will Donald Trump win the 2024 US presidential election? AND Will the men's deadlift record be broken in 2025?'
> because those seem wholly unrelated.
> Can you help me with this? I will just give you two forecasting questions, and you must give me
>
> 1. One or more examples of reasons someone might be interested in the logical combination of those questions; based on how realistic these reason(s) are, provide–
>
> 2. a score between 0 and 10 to advise me on whether it makes sense to consider their logical combination (with 0 being 'the logical combination is nonsensical, nobody would ever ask something like that', 10 being 'yeah that's a perfectly legitimate question I could imagine seeing that on Manifold or Metaculus')

*Figure 9.* The prompt used to decide whether two questions are related enough to be combined in an instantiated tuple.

### E.3. Models and settings used

All settings (incl. temperature) for (Halawi et al., 2024) forecasters are as set by their released code. The temperature in basic forecaster experiments is 0. The versions of the models used are `gpt-4o-2024-05-13` and `gpt-3.5-turbo-0125`.

---

**Verification prompt – CONSEQUENCE**

I will provide you with two propositions, P and Q. Your task is to assess whether Q is a proposition that will always be true if P is true. In other words, validate whether Q is a logical implication of P, ensuring that Q will always occur if P is true. Reject if P and Q are completely equivalent. Q should be a logical consequence of P, but not necessarily the other way around. Reject if you need any additional assumptions to derive Q from P. Reject if Q is just formed by making some resolution criteria more vague / not operationalizing them (but accept if it is made by actually loosening some resolution criteria while still precisely defining everything). Reject if Q is 'ERROR: NO CONSEQUENCE FOUND' or something like that.

Example 1:
P: A computer can receive emails.
Q: A computer is connected to the internet.
reasoning: If a computer can receive emails (P), then it must be connected to the internet (Q), as an internet connection is necessary for receiving emails. Therefore, Q is a logical consequence of P.
valid: True

Example 2:
P: The ground is wet.
Q: It is raining.
reasoning: I can easily imagine the ground being wet (P true) without it raining (Q false). So P does not imply Q.
valid: False

Example 3:
P: It is daytime.
Q: The sun has risen and not set yet.
reasoning: The two statements are logically equivalent, as daytime (P) is defined by the sun being above the horizon and not having set yet (Q). So Q is a logical consequence of P, but also completely equivalent to it, therefore not useful to us.
valid: False

Example 4:
P: Will at least 50 percent of the world's population live in Asia by 2050?
Q: Will Asia have at least 3 billion residents by 2050?
reasoning: They probably thought Q was a logical consequence of P because the world population is 8 billion, half of that is 4 billion, so if Asia has more than 4 billion people it must have more than 3 billion people. However, this assumes that the world population in 2050 is 8 billion, which we do not know for certain. Without knowing the world population in 2050, we cannot judge if 50 percent of that is more or less than 3 billion.
valid: False

Example 5:
P: Will ANY of the following happen in 2025? (a) A manned mission to Mars (b) A new Starship launch by SpaceX?
Q: Will a manned mission to Mars happen in 2025?
reasoning: Suppose only a new starship launch happens, but no manned mission to Mars. Then P is true, but Q is false. So Q is not a logical consequence of P.
valid: False

Example 6:
P: Will there be an epidemic of meningococcal diseases in 2025? (An epidemic is defined as a disease with an infection rate of over 15 per 100,000 people over a span of two consecutive weeks.)
Q: Will there be an epidemic of meningococcal diseases in 2025? (An epidemic is defined as a disease that infects a large number of people over a short span of time.)
reasoning: Q just makes the definition of an epidemic more vague, so we do not count it.
valid: False

Example 7:
P: Will there be an epidemic of meningococcal diseases in 2025? (An epidemic is defined as a disease with an infection rate of over 15 per 100,000 people over a span of two consecutive weeks.)
Q: Will there be an epidemic of meningococcal diseases in 2025? (An epidemic is defined as a disease with an infection rate of over 10 per 100,000 people over a span of two consecutive weeks.)
reasoning: Q is a logical consequence of P, as the definition of an epidemic in Q is a subset of the definition of an epidemic in P. If an epidemic is defined as infecting more than 15 per 100,000 people, it must also be true that it infects more than 10 per 100,000 people, as 15 is more than 10.
valid: True

*Figure 10.* The prompt used to verify whether the generated CONSEQUENCE tuple is logically valid.

# F. Feedback Form Examples

## Example Entry

- **title**: Will the United Kingdom deploy an AI-powered public surveillance system with real-time facial recognition capabilities by 2025?

- **body**: Resolution criteria: This question will resolve as Yes if, by December 31, 2025, the United Kingdom has officially deployed a public surveillance system that utilizes artificial intelligence for real-time facial recognition across any part of its public spaces. The system must be capable of identifying individuals and matching them to a database with at least an 85% accuracy rate. The deployment of such a system must be confirmed by an official government announcement or documentation. In the absence of a public announcement, credible reports from at least three major news organizations (e.g., BBC, The Guardian, Reuters) will be considered sufficient evidence. The system must be operational and not in a trial phase. If multiple systems are deployed, the resolution will consider the first system that meets these criteria. In the event of a partial deployment (e.g., limited to specific cities or areas), the question will resolve as Yes if the system is intended to be expanded nationwide. Edge cases, such as temporary deployments for specific events or the use of similar technology in private spaces, will not count towards this question's resolution.

- **resolution_date**: 2025-12-31 00:00:00+00:00

- **metadata**:

    - **tags**: [United Kingdom]
    - **category**: [Artificial Intelligence]

- **id**: 8da89b38-81d2-43fa-b3ed-4f73a2052c2f

- **Has Previous Feedback**: True

## Example Feedback

- **feedback_id**: 652f3dbd-4935-43d8-8e22-e84d17a1414d

- **question_id**: 8da89b38-81d2-43fa-b3ed-4f73a2052c2f

- **bad_or_irrelevant_included_information**:

- **unintuitive_or_wrong_resolution_criteria**:

- **too_specific_criteria_or_edge_cases**:

- **ambiguities**: Should specify which public news agencies would count as resolution.

- **edge_cases_not_covered**:

- **general_feedback** :

- **formatting_issues**:

- **rewritten_title:**:

- **rewritten_body**: Resolution criteria: This question will resolve as Yes if, by December 31, 2025, the United Kingdom has officially deployed a public surveillance system that utilizes artificial intelligence for real-time facial recognition across any part of its public spaces. The system must be capable of identifying individuals and matching them to a database with at least an 85% accuracy rate. The deployment of such a system must be confirmed by an official government announcement or documentation. In the absence of a public announcement, credible reports from at least three major news organizations (BBC, The Guardian, Reuters, Bloomberg, New York Times, Washington Post) will be considered sufficient evidence. The system must be operational and not in a trial phase. If multiple systems are deployed, the resolution will consider the first system that meets these criteria. In the event of a partial deployment (e.g., limited to specific cities or areas), the question will resolve as Yes if the system is intended to be expanded nationwide. Edge cases, such as temporary deployments for specific events or the use of similar technology in private spaces, will not count towards this question's resolution.

- **rewritten_resolution_date**:

- **discard_reason**:

