# OpenReview forum: "Consistency Checks for Language Model Forecasters"
_ICML.cc/2024/Workshop/Agentic_Markets — Agentic Markets @ ICML'24 Oral_

### Official Review · Reviewer_trZ8 · 2024-06-14
**Paper Review**

**Rating:** 8
**Confidence:** 5

**Review:**

## Summary
This paper introduces an automated evaluation system based on consistency checks for assessing future forecasts generated by large language models (LLMs). The proposed system generates a set of base questions, instantiates the consistency checks from these questions, elicits the predictions of the forecaster, and measures the consistency of the predictions across the questions to detect any violations amongst logically similar questions. The paper introduces two consistency metrics: arbitrage and frequentist metrics to identify logical inconsistencies in predictions. The study highlights the potential applications of this framework for evaluating advanced AI oracle systems, ensuring their reliability and truthfulness.

## Strengths
- The paper presents a novel approach for evaluating LLM forecasters through logical consistency checks, addressing a significant challenge in the field where ground truth for long-term forecasts is unavailable.
- The development of an automated evaluation system with LLM scalable oversight based evaluation generation is a substantial contribution, enabling scalable assessments of LLM-Forecasters.
- The introduction of the arbitrage and frequentist metrics for measuring consistency violations offers a principled and quantifiable method to assess logical inconsistencies or "violations" in probabilistic forecasts. These metrics are well-defined and provide clear criteria for evaluating forecast consistency.
- The detailed pipeline for generating and evaluating consistency checks, from question preparation to consistency scoring, is well-structured and thorough.
- The combination of real prediction market questions with synthetically generated questions enriches the evaluation dataset, ensuring that the consistency checks are both relevant and diverse. Further, synthetic question generation is verified and improved with human feedback, and LLMs are used to filter out question tuples.
- The potential applications of the proposed framework in steering and evaluating superhuman AI oracle systems highlight its practical relevance. The framework could be crucial in ensuring the reliability and safety of AI systems deployed in real-world decision-making scenarios.

## Weaknesses
- The focus on binary forecasting questions limits the scope of the evaluation framework. Many real-world forecasting scenarios involve more complex probability distributions, and the paper does not address how the framework could be adapted for such cases.
- The multiple steps involved, including synthetic question generation, relevance scoring, filtering and manual verification, could be a barrier to adoption. Although synthetic question generation or scoring could be optimised via prefix caching based generation.
- The paper uses OpenAI models, and does not explore the setup with open-sourced models.

---

### Official Review · Reviewer_QU9U · 2024-06-15
**A promising approach for human-independent scalable oversight using consistency checks on forecasting questions**

**Rating:** 8
**Confidence:** 4

**Review:**

This work is an impressive addition to the oversight / oracle AI direction of AI safety and presents a promising approach to identify model mistakes without the explicit oversight from a human. The work is focused on forecasting market consistency to evaluate prediction ability and introduces two new consistency evaluation metrics.

Strengths:
- Theoretically well-grounded work
- Adequate supporting information in the appendix
- Deals with an important oracle AI problem
- Improves and evaluates existing methods

Weaknesses:
- I would like to see more of a deep dive into the median violation arbitrage metric, e.g. a simple histogram. There's some simple questions i.r.t. the distribution of violation.
- Quite complex with a good amount of work but some aspects, such as the human evaluation, could possibly be described or expanded upon

Other comments:
- Anonymity broken at the start of page 4
- Spelling/grammar mistakes